# Current Perspectives on the Challenges of Implementing Assistance Dogs in Human Mental Health Care

**DOI:** 10.3390/vetsci10010062

**Published:** 2023-01-15

**Authors:** Sandra Foltin, Lisa Maria Glenk

**Affiliations:** 1Department of Biology, University of Duisburg-Essen, 45141 Essen, Germany; 2Comparative Medicine, The Interuniversity Messerli Research Institute of the University of Veterinary Medicine Vienna, 1210 Vienna, Austria

**Keywords:** mental health, service dog, assistance dog, emotional support animal, psychiatric dog, PTSD service dog, autism spectrum disorder, post-traumatic stress disorder

## Abstract

**Simple Summary:**

Accounting for the global rise in mental health disorders, sustainable therapeutic strategies are urgently needed. However, despite the increasing interest in dogs that support their owners with a mental illness such as post-traumatic stress disorder, depression or autism, some issues including inconsistent use of terminology and the variability or lack of certification procedures across countries have remained unresolved. Moreover, to date, only few studies have addressed the tasks these dogs accomplish and there is only little information available on the canine welfare status related to the performance in human mental health support. This scoping review stresses the need for stringent procedures in legislation, certification, training of desired tasks and animal welfare management practices. Considering the challenges associated with a mental health diagnosis, collaborations of dog provider organizations and health care professionals would be desirable to continuously assess the efficiency of the human-dog dyad regarding their overall compatibility, general satisfaction and mutual well-being.

**Abstract:**

The prevalence of mental health disorders, driven by current global crises, is notably high. During the past decades, the popularity of dogs assisting humans with a wide spectrum of mental health disorders has significantly increased. Notwithstanding these dogs’ doubtless value, research on their legal status, certification processes, training and management practices, as well as their welfare status, has been scarce. This scoping review highlights that in contrast to other assistance dogs such as guide dogs, there exists no consistent terminology to mark dogs that assist humans with impaired mental health. Legal authorities monitoring the accreditation process, training and tracking of mental health supporting dogs are broadly lacking, with only few exceptions. This review emphasizes the need to address several topics in the promotion of progress in legal and welfare issues related to assistance dogs as well as emotional support dogs for humans with a mental health disorder. The current body of knowledge was assessed in three different areas of focus: (1) the legal dimension including definitions and certification processes; (2) the dimension of performed tasks; and (3) the dog welfare dimension including aspects of the relationship with the handler and risks associated with children recipients. Considering the challenges associated with a mental health diagnosis, collaborations of dog provider organizations and health care professionals would be desirable to continuously assess the efficiency of the human-dog dyad regarding their overall compatibility, general satisfaction and mutual well-being.

## 1. Introduction

The current global crises have cumulative effects and resulted in a significant rise in mental health diseases in recent years [1,2]. Accounting for this emerging trend, appropriate interventions are needed to address the specific needs of mentally vulnerable populations. Assistance or service dogs’ duties have diversified to aid and support people with physical, mental and/or emotional challenges. In the widest sense, they are a versatile group of working dogs that are trained or proficient to assist humans with different types of impairment [3]. The past decade has been characterized by a significant increase in both dogs serving humans with a mental illness [4] and studies that sought to measure the effects on human health outcomes [5]. Notwithstanding these dogs’ doubtless value, research on their legal status including terminology, definition, authorization as well as their repertoire of trained behaviors and overall welfare has been scarce [3]. Apart from assistance dogs, similarly, emotional support dogs (ESDs) seek to positively affect human mental health, although these animals do not receive any training or preparation to fulfil their assigned role [6]. These aspects have contributed to inconsistencies and confusion regarding legal, public and social privileges for the human-dog dyads. 

The aim of this scoping review was to address recent advances and challenges of implementing assistance dogs in human mental health care by analyzing and discussing the state of knowledge on three dimensions: The legal dimension, the dimension of performed tasks and the welfare dimension. Moreover, we sought to elucidate the differences between assistance dogs and ESDs on the respective domains. 

Inclusion criteria for literature were the publication in a peer-reviewed scientific journal, book publications or book chapters, legal documents or statements, position papers or guidelines. Exclusion criteria included duplicate publication, publication as a monograph or academic thesis and language other than English or German. 

Scientific databases (including ScienceDirect, SCOPUS and PubMed), Google and Google Scholar were screened for the following terms: psychiatric assistance dog, psychiatric service dog, emotional support animal or dog, legal status, trained tasks or behavior, performed tasks or behavior, animal or dog welfare. The available literature was screened for relevant information covering the three main areas of interest: the legal dimension, the dimension of performed tasks and the welfare dimension.

## 2. The Legal Dimension

Despite the rapidly increasing numbers of assistance dogs accomplishing various roles, terminology and definition lack clarity and thus impact the question of legal protections given to the owner or handler as well as the dogs. The situation has been complicated in that some jurisdictions afford legal protections also to emotional support animals similar to assistance animals, even though former lack all training requirements [6]. 

### 2.1. Assistance Dog: Terminology, Definition, Numbers, Legal Status

A panel of experts recently addressed the issue of flawed terminology (i.e., assistance or service dog versus emotional support dog) across countries, laws and organizations and provided some recommendations to facilitate working dog classification. Thereby, Howell and colleagues suggest promoting the term assistance dog instead of service dog when referring to a dog that has been trained to perform defined tasks [7]. However, past usage has established the inclusive term service dog [8] in the United States (U.S.), whereas, internationally, “assistance dog” is the most common term [9,10]. Therefore, in this paper, either term, service or assistance dog, are used interchangeably. Terms and definitions are depicted in Table 1. 

Walther et al. [4] researched placement (2013-2014) with persons with disabilities (including mobility, autism spectrum disorder (ASD), psychiatric diseases, diabetes, seizures) on the statistics and varieties of dogs of U.S. and Canadian non-profit organizations of Assistance Dogs International (ADI) and the International Guide Dog Federation (IGDF) besides non-accredited U.S. assistance dog organizations. U.S. and Canadian replying accredited organizations (55 of 96: 57%) assigned 2374 dogs; non-accredited U.S. organizations (22 of 133: 16.5%) consigned 797 dogs. Autism service dogs were the third-highest group allocated by accredited organizations for these two years in the U.S. and Canada (n = 205 dogs) as well as for U.S. non-accredited organizations (n = 72 dogs). The assignment of autism service dogs rose by 16% from 2013 to 2014 in the U.S and Canada for accredited organizations [4]. Autism service dogs particularly support children who were diagnosed with ASD and their caregivers. Psychiatric service dogs were fourth-most common in accredited assignments (n = 119) and accounted for most allocations (n = 526) in non-accredited organizations. The accreditation status of the assistance dog organization was significantly connected with the categories of dogs they assigned. Non-accredited facilities allocated primarily psychiatric service and seizure alert dogs. Accredited organizations often bred their own dogs or used other breeders, but did not utilize clients’ pets or shelter dogs. Non-accredited facilities on the other hand frequently made use of the clients’ own pets or dogs from shelters and did not breed their own dogs. A majority of both facilities train and prepare dogs contingent on the requirements of recipients [4]. A comparable pattern was reported in Europe, but is not represented on an international level, where organizations tend to typically place dogs of only one category [4]. Accordingly, the fastest growing groups (both by accredited and non-accredited organizations) were dogs that assist with autism and psychiatric disabilities. 

ADI [11], an organization linking not-for-profit programs that train and assign assistance dogs, estimates that currently there are 16,766 assistance dogs in the U.S. This number, however, does not take account of assistance dogs trained by their disabled owners. Therefore, it is challenging to establish a precise number of assistance dogs in the U.S. For instance, ShareAmerica.com [12] estimates that there are all in all approximately 500,000 service dogs at work in the U.S., where all states have laws regarding assistance dogs, but individual states differ in their definitions [4]. All breeds and body sizes of dogs are being utilized in assisting roles. In fact, a study of dogs that were registered in California as assistance dogs included equal numbers of large and small dogs, and a lesser number of medium sized dogs [13]. Under the Fair Housing Act (FHA) [14], the law obliges homeowners and housing providers not to discriminate and to make available reasonable accommodation for service dogs. Under the Air Carrier Access Act (ACAA) [15], airline operators in the U.S. are required to accept service dogs as passengers and transport them on flights to, within, and from the U.S. This is exclusive to service dogs and does not apply to emotional support animals and other animal species. The Americans with Disabilities Act (ADA) [16] provides service dogs the right to enter public areas and facilities. Allergies or cynophobia (fear of dogs) are not considered as legitimate reasons for a service dog team to be denied access. The laws safeguard the assistance dog teams, not the assistance dog alone. If the assistance dog is being handled by a secondary handler (such as a parent or caretaker), or (in some states) by a professional trainer, the assistance dog loses the right to be in public places. Legislation and regulations in the U.S. guarantee persons with disabilities the right to have public access with their service dogs that execute tasks related to the recipients’ disability [17]. Although it is mandatory that the dog is trained in these tasks, the method and source of the training are unspecified and no certification process or special identification is obligatory for the dog or the handler [18].

The assistance dogs report published by the European Union Program PROGRESS postulates that comprehensive EU-wide laws are lacking [19]. This deficiency has negative consequences ranging from complications concerning the definition and recognition of the various types of service dogs (e.g., by government institutions) up to individual challenges in the dogs’ and owners’ everyday life, for instance, their freedom of movement in public areas and by using the train or airplane. National laws addressing the diverse types of service dogs are fragmented and vary across EU countries [20]. Currently, the European Standards Agency, TC/452, is attempting to establish an applicable assistance dog standard [7].

Regulations have been implemented by the European Parliament concerning the rights of humans with disabilities when travelling by plane specifying that service dogs may be transported in the cabin; however, it is subject to national regulations [21]. Furthermore, airlines have specific guidelines for the transportation of service dogs which may be at variance, sometimes requiring certification by certain assistance dog provider organizations. British Airways permits service dogs to accompany their owner in the airplane if the dog is certified by a member of ADI or of the International Guide Dog Federation (IGDF) [22]. Lufthansa, a German airline, on the other hand necessitates the dog to be a “recognized” service dog in order to travel with the handler in the cabin free of charge, without restricting the permission to service dogs certified from specific organizations [23].

Austria has implemented laws maintaining requirements and procedures for the official qualification and recognition of assistance dogs. The Messerli Research Institute [24], part of the University of Veterinary Medicine Vienna, was appointed in 2014 as the official coordinating authority for assistance dogs. Requirements and prerequisites to be officially qualified and recognized as an assistance dog in Austria encompass a health and behavioral suitability check as well as a two-step procedure working performance screening [25]. 

In Denmark, service dogs for adults with mental illnesses were first legally recognized in 2012, with emerging numbers of people who request a dog. A case study on the implementation of an assistance dog to relieve symptoms of post-traumatic stress disorder (PTSD) raised the question whether these dogs can be legally considered as helping aids to support daily activities of mentally ill people similarly as for the physically disabled [26]. 

Most countries lack centralized registration processes, thus not requiring any specific accreditation procedure to verify the training and authorization of the dogs and even permitting owners to train their individual assistance dogs. Exceptions are Japan [27], Taiwan [28] Austria [29] and Queensland, Australia [30], where a centralized authority for certifying and tracking assistance dogs exists. In Japan, the Act on Assistance Dogs for Physically Disabled Persons was established in 2002 [31] with the aim to facilitate the quality of assistance dogs and the use of public facilities for people with physical disabilities. It guarantees patient and dog open access as facilities cannot deny access to certified guide, service and hearing dogs. In Japan, the term “assistance dogs” refers to guide dogs, mobility service dogs and hearing dogs certified in accordance with the Act [31]. All other dogs are categorized as “pets”. The Act specifies the rights and duties of assistance dogs training organizations, certifying organizations, and eligible assistance dog partners. In 2019, Japan had 26 training organizations for service dogs [32] and seven organizations certifying service dogs. Sixty-six service dogs were registered in 2019 [33]. In Japan, service dogs are solely used for people with physical and not emotional or psychological disorders. The service dogs are lent to the handler by a training organization gratuitously. The handler is responsible for the health and well-being of the dog, which may become challenging depending on his or her medical condition [32]. Once the dog has completed its training and passed the examination, it becomes an official and registered service dog. The training organization remains responsible for continuing training and assistance and is obligated to provide re-training in the event of changing circumstances [34].

### 2.2. Emotional Support Dog (ESD): Terminology, Definition, Numbers, Legal Status

According to Service Dog Certifications (SDC) [35], an ESD is a pet dog to offer a health benefit and/or support for those that suffer from an emotional or mental disability (see Table 1). ESDs are used for an extensive range of mental illnesses such as Attention Deficit Disorder; Learning Disorders; ASD; General Anxiety Disorder; Gender Identity; Bipolar; Cognitive disorders; Depression; Severe anxiety and/or PTSD. The Americans with Disabilities Act (ADA) does not necessitate an ESD registration [14]. The number of ESDs has similarly increased significantly within the last years [36]. ESDs in the widest sense are dogs that offer some type of companionship or support that will aid alleviate at least one aspect of their owner’s disability [7]. The U.S. Centers for Disease Control and Prevention estimate that in 2019, there were nearly 200.000 emotional support animals (ESA) in the U.S. [37].

At present, a valid letter signed by a qualified, licensed healthcare provider is prerequisite to have a dog legally designated as an ESD for the purpose of housing accommodations [8]. The ESD designation does not however grant public access in any other contexts. The dog must be of importance to assist in a person’s daily functioning. No special training or suitability screening is obligatory [6,38]. 

The lack of standards, procedures and certifications regarding ESDs has resulted in much confusion [7,39], fostering a market for falsely “certifying” pets. This led to a bill in in the state of Michigan that will penalize people who sell online certificates [40]. The American Disabilities Act and Department of Justice do not recognize any form of certificate or identification card as a proof of an animal’s designation as a service animal or an ESD [41]. A “registration” or “certification” does not constitute appropriate documentation of any kind of helper animal [40].

**Table 1 vetsci-10-00062-t001:** Assistance Dogs and Emotional Support Dogs (ESD): Terminology, definitions, recipient.

	Assistance Dog/Service Dog [7,14]	Emotional Support Dog (ESD) [7,16]
Definition	“An animal living with and highly trained to mitigate the impacts of the owner’s disability, and with legal protections” ([7], p.3) The ADA defines service dogs as “dogs that are individually trained to do work or perform tasks for people with disabilities” [14] The U.S. Department of Justice uses the following definition: “Service animals are defined as dogs that are individually trained to do work or perform tasks for people with disabilities. Service animals are working animals, not pets. The work or task a dog has been trained to provide must be directly related to the person’s disability. Dogs whose sole function is to provide comfort or emotional support do not qualify as service animals under the ADA” [15]. “an animal who performs at least one identifiable task or behavior (not including any form of protection, comfort, or personal defense) to help a person with a disability to mitigate the impacts of that disability, and who is trained to a high standard of behavior and hygiene appropriate to access public spaces that are prohibited to most animals” ([7], p.6)	An emotional support animal (ESA) may be an animal of any species (domestic, rare or exotic) that provides some emotional or therapeutic support to an individual with a mental health condition or emotional disorder [20]. “an animal who lives with and provides emotional benefit and/or support for the person, as confirmed by an appropriate qualified health care professional.” ([7], p.7)
Recipient	Individuals with a physical or psychological disability as defined by the Americans with Disabilities Act (ADA).	Any individual whose need is expressed in a request by a qualified physician, psychiatrist, or other mental health professional based upon a disability-related need.

### 2.3. Organizations and Health Care Providers

Ethical principles and codes of conduct state that mental health clinicians provide only services within the realm of their competency. When offering novel services, clinicians have to take on pertinent education, training or study. The American Psychological Association (APA) Ethics Code 2.01b states that “psychologists have or obtain the training, experience, consultation, or supervision necessary to ensure the competence of their services, or they make appropriate referrals…(c) Psychologists planning to provide services, teach or conduct research involving populations, areas, …new to them undertake relevant education, training, supervised experience, consultation, or study” (American Psychological Association, 2017, paras. 2–3). The American Counseling Association (ACA), 2014, p. 8 states: “While developing skills in new specialty areas, counselors take steps to ensure the competence of their work and protect others from possible harm” [42].

Mental health providers should consequently have comprehensive knowledge on the subject of therapeutic human-dog interactions, canine behavior, and a thorough understanding of the policies surrounding ESDs at the local, state and federal level. Without this knowledge, clinicians, regardless of discipline, risk practicing outside their scope of competence. Regarding ESD prescription, mental health providers should consult or cooperate with dog trainers, behaviorists, veterinarians and/or providers of animal-assisted interventions. Given the absence of accrediting organizations and of federal or state mandates, the mental health provider has to ensure ethical practices43]. The mental health professional must conduct a thorough assessment of a person applying for an ESD to define disability-related need [43]. Prior to endorsing an ESD, the mental health provider has to make reasonable efforts to ensure that the client can provide adequate food, water, housing and veterinary care and fulfill the emotional requirements of the dog, despite their disability. Illness, undue stress from being handled by a person or persons without specialized training in animal welfare, or injury from interactions with the public have to be discussed and minimized.

It is critical that organizations and health care providers have extensive expertise and knowledge regarding a future dog handler’s specific disability, cognitive ability, the types of prescribed medication and consumption of addictive substances as well as the effect of comorbid conditions prior to dog adoption. Expectations of first-time handlers regarding the dog must be evaluated. Unrealistic expectations should be minimized as they negatively impact perceived success and satisfaction with the dog. Organizations and health care providers must delineate resources required for successful integration of a dog into the recipient’s life. This involves several aspects. Does the handler receive extensive ongoing support? Do the providers maintain contact with handlers with multifaceted disabilities? Do organizations and health care providers have the resources to attend to complex cases? Are alternative placement models an option, such as offering initial training prior to dog adoption, particularly for individuals with complex disabilities and cognitive impairment and/or additional support throughout the assignment duration, perhaps for the entire lifetime of the dog?

### 2.4. Outlook

Future legislation should aim at addressing fundamental aspects such as providing internationally consistent definitions of the various types of service dogs or ESD for people with mental health disorders; the implementation of qualification procedures for these individuals; and the establishment of an official registration system to provide transparency for government institutions with regard to dog and human data (number of dogs, distribution, age, breed, assessment results, etc.). This database could moreover offer the option to track individual dogs, to assess their welfare state, and to provide dog owners with the opportunity to regularly participate in educational initiatives in order to improve their individual knowledge about dog training and welfare maintenance.

## 3. The Dimension of Performed Tasks

### 3.1. Psychiatric Assistance Dog

As indicated in Table 2, psychiatric assistance dogs (PAD) are trained to provide disability-specific support to one person (i.e., the recipient) [44,45]. The tasks these dogs are proficient to perform typically include physical chores [46], and emotional, social and psychological benefits [47] to increase the recipient’s well-being [48] and quality of life [49]. According to findings by Lloyd et al. [50], psychiatric conditions in Australian PAD handlers most commonly included depression (84%), anxiety (social 61%; generalized 60%), PTSD (62%) and panic attacks (57%), while fewer study respondents suffered from Obsessive-Compulsive Disorder, ASD and eating disorders. 

The most common associated tasks performed by the assistance dogs were anxiety reduction through tactile stimulation (94%); nudging or pawing to disrupt dissociative states (71%); interrupting an undesirable behavioral state (51%); maintaining constant body contact (50%); deep pressure stimulation (45%); and blocking contact from other people (42%) [50]. Other performed tasks reported by Lloyd et al. [50] included alerting the recipient to leave the bed or house; reminding the recipient to take medications; providing safety; sensing recipient’s emotions and behaviors; and providing a “reality check” from anxiety or dissociation/hallucination. While no statistically significant associations emerged between the recipients’ mental health diagnoses and the tasks the dog performed, reductions in the consumption of mental health care services were caused by fewer suicide attempts, less hospitalizations and less medical requirements. However, in 54% of the study participants, having a PAD was not linked to a lesser need for seeking psychiatric or mental health care. 

According to Tseng [51], assistance dogs may help to alert and/or interrupt potentially problematic repetitive or self-stimulating behaviors in children with ASD. In addition, dogs can apply pressure stimulation to the children that resembles patterns of touch therapy that are commonly practiced by occupational therapists with the aim of alleviating arousal and anxiety [51,52]. Dogs may provide a calming presence and decrease the numbers of disruptive behaviors including tantrums [53,54]. In addition, parents with an assistance dog perceived the public to react more responsibly and respectfully towards their children [54]. Moreover, reductions of child salivary cortisol levels have been related to the presence of an autism assistance dog [55]. A recent assessment of chronic cortisol in hair or nail specimen points at reduction in cortisol levels for both parents and children. Reduced chronic cortisol concentrations were further paralleled by reductions in stress levels as perceived by the parents [51].

To prevent an autistic child from elopement, a special belt connecting the child to the dog’s vest has been previously used. While a parent or adult may hold the dog’s lead, the performed behavior of the dog is to passively resist with its body weight or lay down if the child attempts to escape, thereby securing it in public spaces or near roads [52,53,54,55].

### 3.2. PTSD Service Dog for Veterans

Up to 23% of the United States military personnel deployed to Iraq and Afghanistan returned with diagnosed symptoms of PTSD [56]. PTSD has been defined by the APA by avoidance, re-experiencing traumatic events, impaired cognition and mood as well as agitation [57]. Symptom alleviations in terms of behavioral adjustments are anticipated from the support of the PTSD service dog. PTSD service dogs for veterans are supposed to alert to and interrupt anxiety and panic attacks and are frequently trained for positional commands such as standing behind the veteran in public and “watching their back”, thereby providing a sense of safety [58]. 

The demand for PTSD service dogs is immense and it may take months or years to acquire a dog [4,59]. One reason for the extraordinary demand may be that PTSD service dogs are not as publicly stigmatized in contrast to other mental health treatment options [60]. Empirical assessments of the specifically trained tasks the dog is supposed to perform for military veterans with PTSD, such as the therapeutic components of the intervention, remain mostly undefined. Several proposed criterions for PTSD service dog training propose that the dogs must be able to lessen the veteran’s PTSD symptoms [61]. These trained assignments, however, vary widely across service dog providers, are not specified and have to be applied to a veteran’s individual requirements [62]. Accordingly, no assessments exist on how significant untrained versus trained behaviors are for decreasing PTSD symptoms.

Rodriguez et al. [63] assessed the relevance of trained and untrained behaviors of PTSD service dogs regarding their relevance, frequency of use and PTSD-symptom specificity for post-9/11 veterans. Research findings suggest that calming and interrupting anxiety were considered as the most important and most frequently used trained tasks to mitigate symptoms of PTSD. Veterans who had an intense relationship with their dog relied on trained tasks more often and veterans who had their service dogs longer less frequently required trained tasks. Cover and interrupt/alert to anxiety were considered as the second- and third-most important behaviors [63]. Other symptoms helped by the service dog were decreasing intrusive memories of a traumatic event, feeling distressed and having intense bodily responses (e.g., palpitations and sweating). Veterans on the waitlist had higher expectations compared to veterans with a service dog. The most relevant untrained behavior for relieving PTSD-related symptoms was to express love for the dog and to feel loved in return, while the least important task was to provide social help in public [63].

Other studies similarly propose that anxiety-reducing behaviors displayed by service dogs are the most essential and key mechanistic components to veterans for reducing hyperarousal and dealing with re-experiencing events [58,59,64,65,66]. The cover task was the second-most frequently used and is thought to replicate aspects of military comradeship by “watch my back” missions in which soldiers will guard each other during combat. Veterans described that their service dogs, similarly to themselves, are constantly on alert or aware of approaching people [58]. When performing “wake up from nightmare” task, the dog responds to stress indicators at night and interrupts the troubled sleep episode of the veteran. The dogs were to reduce the symptoms of intrusive memories or flashbacks as well as internal and physical distress by the trained tasks of “calm/comfort to anxiety” [59]. Williamson et al. [67] integrated the multifaceted support provided by a PTSD service dog into the four principal components of the Zooeyia model, thus labeling the service dog as a builder of social capital, agent of harm reduction, motivator for health-related behavior adaption and active participant in healthcare. 

### 3.3. Emotional Support Dog (ESD)

Given that ESDs do not require any training, only few data exist on the tasks they perform. As shown in Table 2, ESDs provide emotional support, thereby alleviating specific symptoms or disability by their mere presence [68]. To this end, ESDs primarily provide companionship, relieve loneliness, and sometimes contribute to improve symptoms of depression, anxiety, and certain phobias. ESD are used in various heterogeneous environments for emotional support, including rides on airplanes, being taken to educational or working settings [69] or assisting during daily activities such as shopping without having had any prior training. 

Brooks et al. [70] reviewed 17 studies regarding ESAs. Of the 17 studies, eight were conducted in the USA, four in the UK, two in Canada, one in the Netherlands, one in Australia and one in Sweden. Fifteen reported positive features of dog ownership for individuals suffering mental health problems whereas nine described negative elements. In military veterans with PTSD, positive effects were centered on decreasing feelings of loneliness, depression, worry and irritability, and, moreover, having a source of comfort and affection [71]. Owner described that their dogs provided a source of physical warmth and companionship, opportunities for communication, reducing feelings of isolation, provided a safe environment and “their dogs allowed them to express their feelings and clarify their thoughts without the concern that they will interrupt, offer criticism or advice or betray confidence” [72]. The studies further indicated that the dogs provided unconditional love and affection fostering self-acceptance and congruence, supporting emotional stability, aiding in stress management and coping with challenging life events [73]. For people living alone, dogs were a source of “connectedness” [74], reassurance, and normalcy [75]. The dogs contributed to their owner’s feelings of security by distracting from particular symptoms such as hearing voices or panic attacks and through decreasing symptoms of their mental health disorder [70,71,75,76].

However, a recent survey study failed to expose any model for evaluating individuals reporting need for an ESA [77], outlining the general deficiency of awareness regarding the ESA policy of many mental health professionals and the absence of reliable assessment standards.

### 3.4. Training Methods

Employing exclusively reward-based (positive reinforcement) training methods is more effective than aversive, compulsive, punishment-based (e.g., shock collars) or mixed methods [78,79]. Solely utilizing positive reinforcement was linked to more optimistic dogs, who learn faster and exhibit more consistent behavioral responses, less pain and suffering as well as lower incidences of negative affect, aggression and problematic behaviors (e.g., unwanted barking) [79,80]. In many instances, the working equipment, such as collars, leads and harnesses, is outdated [81]. Preliminary research considering the trained behaviors of service dogs has mainly focused on assessments to advance the selection and performance to enhance program success rates, which are estimated approximately 50% across different working dog populations [82]. Assessments of behavioral characteristics should be measured regarding their prognostic value linked to working suitability [83]; the genetics of service dog behavior [84]; maternal care in service dog breeding programs [85]; and development and testing of cognitive skills [86]. Terminology used to designate behavior however differs extensively between and across industry sectors, potentially producing irritations for researchers, dog trainers and dog handlers/recipients alike [82]. Further research is essential to debate the difference and value of untrained versus trained behaviors of the service dog as they may dually contribute to the therapeutic efficacy. 

## 4. The Welfare Dimension 

Animal welfare refers to the quality of life experienced by the animal, positive or negative, with respect to the domains of nutrition, environment, physical health, behavioral interactions and mental health state of the animal [87]. The Five Domains Model encourages the requirement to provide opportunities within each domain that lead to positive affective states [87]. Modern animal management practices are critical to industries reliant on animal use, including working dogs [88]. Risk assessment to safeguard the welfare of assistance dogs at the operational level recognizes a widespread lack of transparency, stakeholder commitment and partaking of best practice and standards [82]. Acknowledgment that dogs are sentient animals, having intrinsic value beyond their consideration as properties, equipment or working aid has been somewhat mirrored in advances to legislation and politics globally (e.g., Australia, European Union, New Zealand, Canada, United States and United Kingdom) [89,90,91]. Service or assistance dogs are permanently housed with individuals with disabilities to aid in their day-to-day life [5] and their welfare must be a principal consideration moving toward a greater degree of concern [92,93]. Concerns such as absence of a day-to-day routine, lack of sufficient “time off”, being overweight, and (un)intentional harm and mishandling of the dog by recipients/owners must be addressed. To this end, provider organizations necessitate transparent, long-term procedures and processes for animal welfare [94,95]. 

Cultural difference clearly exists with respect to the perception of assistance dogs. An Australian study [96] researched the Australian public’s opinion toward both assistance and companion dogs. There was an equal interest in both groups of dogs. Assistance dogs were rated as happier than companion dogs by the study respondents. The survey revealed some ethical concerns regarding the use of animals for human benefit. These included conditions in which the dogs might not profit from their role as working animals and the prevalence of inappropriate behaviors from strangers in public environments such as touching the dog while working. There was only little disapproval of the intensity of training imposed on the dogs as well as the straining and restrictive nature of assistance dog work. It emerged that some study respondents expressed some concern about the ability of recipients with disability to appropriately take care for their dogs, while others stated that assistance dogs may even be better cared for because of the recipients’ dependency on them and the greater amount of time spent together [96]. The past decade has been associated with some advancement in considering the welfare of assistance dogs [82,87,97]. Numerous researchers in animal welfare science have stressed that professionals working with service or therapy dogs (and other animals) must consider welfare issues [92,97,98]. It is vital to understand that if the working experiences repeatedly cause discomfort, pain or fear the dog is inevitably subject to distress which has severe consequence for both its physical and mental health [97]. 

The “ethics of care” approach endeavors to respect all parties participating by emphasizing sustainable relationships and establishing a human-dog bond [99]. Within this model, the dog should be provided with an optimal rather than a sufficient quality of life [82]. The “rights approach” concentrates on protecting and respecting the rights of all individuals involved and furthermore contemplates the advantages and disadvantages from the dog’s perspective. According to the “utilitarian approach”, a cost benefit analysis must determine what should be acted upon subsequently, founded on all the morally relevant consequences (typically harms and benefits for sentient individuals) [100].

### 4.1. Relationship and Attachment

After adoption, guide dogs were broadly considered as family members by their owners [101] and similarly, service or assistance dogs frequently have multiple handlers. Dogs perform differently for respective handlers [102] based on the interplay of canine and human personalities, the strength and style of attachment [103] and recipient beliefs [104]. A majority of recipients experience mental health challenges which may influence their concentration, fatigue, stress, motivation and determination. Some recipients may have extended hospital admissions, not being able to care for their dog during that time span. Recipients with an intellectual disability or a lack of maturity may experience more challenges related to memory and consistency in handling the dog and, thus, impacting its welfare.

Therapy dogs are most commonly visiting dogs that are temporarily brought to a setting to support individuals diagnosed with a mental health illness or to interact with people with other diagnoses, for recreation purposes or social facilitation. They are usually accompanied by their owner or a closely familiar person who acts as the responsible person to safeguard the dog’s welfare at any time [105]. In contrast, mental health service dogs are constantly paired with their owners who themselves frequently have trouble to regulate their inner tension and mental states. Individuals with impaired mental health may experience extended periods of dissociation, where they do not at all respond to their social environment. However, similar to a child, a dog requires attention, social interaction, food, water and needs to be taken outside irrespective of the current constitution of the handler. And the question that instantly emerges is whether people who struggle to maintain their own well-being can be the responsible person to safeguard their dog’s welfare and provide a safe haven in case the dog needs emotional support. 

Measurements of cortisol as a strategy to monitor the adrenocortical activity in response to stress have been broadly applied in canine welfare science [105]. In comparison to saliva, in which cortisol typically rises in response to an acute stressor, analyses of hair may represent the long-term cortisol status [106]. Research by Sundman et al. [107] demonstrated significant interspecies associations in long-term stress when dog hair cortisol levels were correlated with human hair cortisol concentrations. This is particularly interesting since cortisol in hair of PTSD service dogs for veterans did not differ from concentrations of regular pet dogs, suggesting no profound effect of their working task on their long-term stress profile [108,109]. In line with these findings, van Houtert et al. [109] also stated that salivary cortisol, a representative of acute arousal, appeared to be lower in PTSD dogs for veterans during training sequences than concentrations measured upon arrival at the training site or after an episode of free play. According to these recent studies on adrenocortical activity related to supporting veterans with PTSD, the data do not raise any concern about compromised animal welfare. Although the two recent studies have not found any evidence of altered stress levels, both acute and long-term, in PTSD service dogs for veterans [108,109] it would be essential to run long-term research including regular, continuous physiological and psychological follow-ups. As we have mentioned before in therapy dogs [97], studies would benefit from adding measurements of the recipients’ emotional competence, attitude and empathy towards their dog. 

Yarborough et al. [60] described difficulties in coping with the additional stressors of sustaining the dog’s training, integrating the dog into the family, and receiving unwanted attention in public which may cause added stress, anxiety and fatigue. Additionally, findings from a recent exploratory study by Williamson et al. [67] point at difficulties such as establishing a sustainable emotional and working relationship with the service dog as well as integrating it into the family (including socialization with preexisting pets). In case of unsuccessful matches, dogs may need to be rehomed or reassigned to another veteran. Furthermore, preexistent pet dogs may not necessarily be ideal candidates for PTSD service due to challenges of coping with the newly assigned tasks, as Williamson et al. revealed [69]. 

According to Glintborg and Hansen [26] people with PTSD can benefit from their service dog through ventriloquizing (i.e., talking about the nonverbal animal by implicitly referring to oneself) which may facilitate social interactions and psychotherapeutic processes. However, the fact that people with a mental health disorder may regularly have their service dog accompany them to psychotherapy, especially in group sessions, necessitates attention from an animal welfare point of view due to interindividual emotional contagion of behavior and physiology [110] that may negatively impact the dog [111]. 

### 4.2. Children Recipients

Dog assistance may benefit children affected by neuro-developmental disorders such as ASD [112,113,114]. However, families may have unrealistic expectations of the recipient/owner-dog relationship [115], and insufficient information regarding the impact on the dog and the basic welfare guidelines to protect the animal [116]. 

Living in close proximity with children may negatively affect dog well-being and quality of life [117,118], conceivably augmenting the risk of child-directed aggression. The majority of dog bite accidents (about 75%) occur in the household environment [119] indicating a great necessity to intensify parent sentience about contexts and child activities that may trigger a dog bite [120]. Dogs do not necessarily enjoy being physically close to and tactile with a child [121]. Studies suggest that dogs may well find close interactions such as kissing or petting stressful, as evidenced by increases in cortisol [122] and behavioral indices [123,124]. 

A number of studies provide data on the impact of child-dog interactions for dog quality of life [125,126,127], with one considering the quality of life of pet dogs around children [127]. Oher studies referred to therapy dogs [128,129], other animal-assisted intervention dogs [126] and autism service dogs [53]. 

Four papers specified possible sources of stress (for service dogs, therapy dogs and pet dogs), either as identified through the parent/handler [53,127] or through researcher observations [53,129]. 

Hall et al. [127] as well as Burrows et al. [53] reported that child meltdowns and tantrums were particularly stressful for pet dogs [127] or assistance dogs [53]. Stress indicators observed were barking, jumping up and shaking. Both papers indicated that the dog was being at risk from potentially aggressive behaviors from the child, either because the dog was the closest target for the child or because the parent had encouraged the dog to interrupt the display of aggression to calm the child. At times, Hall et al. [127] observed that the dog spontaneously interjected in a meltdown by seeking physical proximity with the child, reflecting its efforts to appease a stressful situation and defuse a perceived conflict [130]. 

Not all negative attention directed to the dog was the consequence of meltdowns or tantrums, as it was reported that the child jumped, prodded and poked the dog aggressively and in a rough manner during various daily interactions [53,127]. 

Lack of predictable routines, uninterrupted resting times, children handling the dog on the lead [127], environmental instability, erratic unpredictable movements, such as those involving wheeled toys or the child bouncing around, as well as loud noises [53,127] and activities from child visitors [127] were considered as major sources of stress for dogs. Recreational activities, in particular sleep and off-duty time, were seen as vital to ensure physical and psychological health [53]. Children’s toys and games had a negative effect on some dog’s well-being as indicated by a proliferation in stress behaviors associated with avoidance (e.g., pet dogs hiding and running away [127]). Interestingly, parents believed that the dog enjoyed being “dressed-up” by the child [127]; however, other studies indicated more stress when children put a bandana on the dog [129]. Behaviors displayed by the dog during child-dog interactions included paw chewing, lip-licking, grooming, yawning and panting [53,127,128,129], running away, shaking, urinating and defecating [53,127], safety seeking behaviors (hiding, going to their safe place or seeking the parent/handler), all of which may be indicators of heightened stress levels [131]. Dogs living full time with children displayed a spectrum of physical health conditions associated with chronic stress due to impaired immunity [119], such as ear, eye and skin infections [53,127]. Social effects exhibited by the dog were displaying distress when left alone with the child shown by whining, scratching, seeking behavior, particularly if the dog had to sleep in the same room as the child [53], or if the child created stressful situations for the dog [127], with the dog also withdrawing from the child [127].

Arousal-reducing interactions that comfort the child (e.g., tactile stimulation from the dog lying on the child, fiddling with the dog’s ears and medallion, and being able to rub the feet on their dog) as reported by Appleby et al. [52] may cause considerable discomfort in the dog, especially if performed for extended periods of time. For appropriate supervision of children and dogs, parents must be aware of the dog’s communication and signaling as reaction to their own or their children’s interactions. 

Parents of neuro-typically developing children and children with neuro-developmental disorders identified various behaviors that pose a risk to the dog. Parent surveys also indicated that children were often not sufficiently supervised around their dogs. Even though dog-child interactions did not differ much between families, study findings revealed that neuro-typically developing children displayed fewer outbursts of anger, crying, meltdowns and tantrums compared to children with a neuro-developmental disorder [127]. The shorter the time the dog had lived with the child, the greater the score on canine excitability in meltdowns and with child visitors. Scores on fearfulness were significantly higher when the child-dog interaction included physical interactions with the dog, such as rough contact, meltdowns and grooming/bathing. Moreover, smaller dogs, older dogs and younger children were linked with increased levels of fearfulness. 

Displays of acute and chronic stress [132] include, e.g., chewing, lip-licking, yawning and panting, cowering, shaking, running away and aggressive behaviors. Another point of concern is that children and adults had difficulty recognizing signs of stress in their dog [121,133,134]. Hall et al. [127] found that the main caregivers tended to report lower signs of fearfulness-related behavior of the dog either indicating desensitization with repeat exposure, or reduced observation of the behavior because they did not recognize relevant behaviors and/or they were unwilling to report them.

If the dog lives full time with the child, the duration of child-dog contact time and the nature of exposure the dog experiences have been considered critical [53,127]. Thus, increased precautions must be taken when an assistance dog or ESD is first placed into a family to support a child diagnosed with ASD.

### 4.3. Physical and Mental Health

The physical health of a service dog must be considered from the time of birth or recruitment, throughout the dog’s working life and into retirement [95], requiring standards for breeding, rearing and/or recruitment of the dogs; housing, transportation; training techniques and equipment; trainer and owner/handler education; enrichment, maintenance of a good welfare state, as well as animal agency [135]. Older dogs naturally become less resilient, need more time to recover from stress, and thus, may not be as adept at managing social situations [136,137]. However, there are no data available about the ideal age for a dog to start working or when to retire [97]. 

Preventive veterinary care is critical to maintain service dog health and an example of minimum requirements are described in the AAHA (American Animal Hospital Association) recommendations [138]. Physical fitness is an essential welfare consideration [87]. Research and rethinking of the importance of veterinarians in detecting and reporting animal abuse and cruelty in the context of service or assistance as well as emotional support dogs is needed. Optimal rest and sleep are critical for all but especially for working dogs. Sleep is connected with emotional states and essential for learning, immune function, performance and recovery [139,140]. It is furthermore critical that service dogs’ social and emotional needs are met by providing them social, environmental and mental enrichment [141] and by allowing dogs to engage freely with their environment under their own motivation. This is referred to as agency [142] and promotes behavioral variety and welfare [143]. Supporting regular occasions for service dogs to exercise agency in environmental and social contexts is vital.

One activity shown to induce positive judgement bias in dogs is nosework [144]. Allowing dogs to engage in olfactory-based sniffing activities resulted in them exercising autonomy and increased optimism [144]. Behavioral problems are a key contributing factor to the extraordinary failure rates in service dog programs [145]. Improved assessment and personalized maintenance for dogs ensure welfare benefits [83]. Community stances and media debate have encouraged modifications in some segments that historically euthanized working dogs as an end point to their training or working life [e.g., Royal Australian Air Force Wilson: [146]; US Military: Alger and Alger [147,148]]. The identification of behavioral displays of affective state and welfare regarding training, operational environments, kennel facilities and home settings should be focused on in-depth [149]. 

A One Health framework [117,150] for the spectrum of dogs supporting human mental health, outlining under which circumstances no tradeoff of human benefits against animal health and well-being may be found, is essential. Assistance dogs must no longer be perceived as objects other than pet dogs being instrumentalized to fulfill a plethora of tasks. As dogs do not share equal privilege, right or power as the recipient, it is questionable whether dogs may give any form of free and informed consent to fulfill the numerous tasks assigned to them [97]. Current perspectives on animal status denote that latter should not be seen as “less than” or “tools” but as individuals with likes, dislikes and limitations [151]. If we maintain the concept of animals as sentient beings whose rights and dignity may be inseparable from our own, then the ethical principles of autonomy, beneficence, nonmaleficence and justice ought to be applied [152].

## 5. Conclusions and Future Directions

During the past decades, the popularity of dogs assisting humans with various mental health disorders has significantly increased. This article highlights that in contrast to other assistance dogs such as guide dogs, there exists no consistent terminology to mark dogs that assist humans with impaired mental health. Legal authorities monitoring the accreditation process, training and tracking of mental health supporting dogs are broadly lacking, with only few exceptions. To advance the field of working dogs in mental health, stringent procedures in safeguarding dog welfare need to be established. Considering the challenges associated with a mental health diagnosis, collaborations of dog provider organizations and health care professionals would be desirable to continuously assess the efficiency of the human-dog dyad regarding their overall compatibility, general satisfaction and mutual well-being. 

A limitation of this scoping review is the relatively narrow focus on specific topics rather than a holistic synthesis of the literature and internet-based content. One facet that has merely received scarce attention is the efficiency and appropriateness of training and management practices. Future research is challenged to raise methodological rigor and uniformity in terminology and definitions. Dog selection processes and the definition of desired traits and performed tasks need to be transparent, allowing a comprehensive evaluation of research. This refers to both assistance dogs and ESDs that are paired with people who suffer from a mental health disorder. Future studies must also determine appropriate and individualized parameters and feasible methods to investigate the ideal life phase for dog retirement and to prepare the recipient for such. Accounting for the variations in which dogs are working within the assistance or emotional support context, comprehensive, long-term studies using objective and multiple measures of welfare are warranted. Methods to measure the relative resilience of dogs to stressful events and the development of optimal protocols to enhance such resilience are certainly needed.

## Figures and Tables

**Table 2 vetsci-10-00062-t002:** Assistance Dogs and Emotional Support Dogs (ESD): Performed tasks and training.

	Assistance Dog/ Service Dog [7,14]	Emotional Support Dog (ESD) [7,14]
Performed tasks	Dogs perform at least one specific assistive task for individuals with a physical or psychological disability as defined by the Americans with Disabilities Act (ADA). Disability-mitigating tasks include the following: Opening and closing doors; turning light switches off and on; barking to indicate that help is needed; providing deep pressure; pulling a wheelchair; alerting to a medical crisis; providing assistance in a medical crisis; grounding their handler during a flashback; guiding their handler home during a dissociative episode; initiating tactile intervention when a handler experiences sensory overload; alleviating symptoms of hypervigilance.	ESD provide companionship, affection and support to people diagnosed with mental and emotional disabilities, autism, anxiety and panic attacks, depression and various phobias. They provide comfort or emotional support by their mere presence [6].
Training	Service dogs are usually professionally trained but may also be trained by their owner. A service animal must be under the control of its handler. Under the ADA, service animals must be harnessed, leashed or tethered, unless the individual’s disability prevents using these devices or these devices interfere with the service animal’s safe, effective performance of tasks. In that case, the individual must maintain control of the animal through voice, signal, or other effective controls [13].	ESDs do not require specific training, licensing, registration, or certification and do not have to be trained for any particular task [14,7].

## Data Availability

Not applicable.

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
