# Peer review of "Current Perspectives on the Challenges of Implementing Assistance Dogs in Human Mental Health Care"

_vetsci, 2023, doi:10.3390/vetsci10010062_

Round 1

Reviewer 1 Report

Thank you for giving me the opportunity to read the manuscript. This is a very useful topic for any readers who are involved in examining human-animal bonds with a special focus on animal-assisted interventions. As a clinical psychologist, involvement in clinical services has received a rapid increase in attention among the profession and potential service users. 

Having said that, although this is a very informative manuscript since this is a manuscript (not a book chapter) submitted to an academic journal, I would appreciate a more research paper structure in presenting the useful information. 

If this is not a systematic review, a scoping review, not a narrative review, please state so what type of review the authors have adopted. This links to the assessment of the scope, depth, and quality of the included information in the manuscript. Although there are some mentionings about the situation in Asia, it was very very thin. The structure of the whole manuscript should be improved. For instance, the Table was introduced very early in the manuscript which I think, could be inserted in a much more appropriate way that can relate back to the content; there, I believe, was a typo in the title of the manuscript. There is no summary of the manuscript nor a limitation of the review. As I compared this with a book chapter, it may be useful if the authors could make a reference to a book chapter published in BSAVA Manual of Practice Veterinary Welfare (https://www.bsavalibrary.com/content/chapter/10.22233/9781910443798.chap9) to see if addition can be included in the manuscript. The manuscript is very timely and useful if the information can be presented much more structurally. 

Author Response

Reviewer 1

We appreciate the overall positive evaluation of the content of our manuscript and are pleased that the reviewer finds the topic interesting and timely. We are thankful for the careful evaluation of our manuscript. Your suggestions have been incorporated and helped us a lot to facilitate reading and to enhance the quality of our paper.

We agree that the structure of our initial submission has needed some rework and therefore, we re-structured the whole manuscript. An introduction has been added and the table has been shortened and moved. We added that this is a scoping review in the introduction. More information on Asian legislation was added. To avoid repetitions and redundancy, many sections were shortened. The suggested reference (https://www.bsavalibrary.com/content/chapter/10.22233/9781910443798.chap9) was included where we found it appropriate. We added a section on conclusions and future directions and addressed the limitations.

Reviewer 2 Report

The Authors submitted their manuscript as Article, however this is not an original research paper. Thus, it has to be moved to a different category: likely Review or Commentary depending on the aim. The aim is not defined, this has to be rectified too. Without it, it is impossible to evaluate the manuscript. The chapters switch between topics and countries with no consistency, e.g. first it is said that the U.S. has no system for monitoring the numbers or types of assistance dogs while in contrast Japan, Taiwan and Austria have a centralized authority for certifying and tracking assistance dogs. However, the numbers of assistance dogs are given for the U.S. and the others.

Author Response

Reviewer 2

We are thankful for the evaluation of our manuscript. Your preliminary suggestions have been incorporated and helped to facilitate reading and to enhance the quality of our paper.

Unfortunately, the paper was first accidently submitted under the wrong type of submission and has now been classified as a review. We agree that the structure of our initial submission has needed some rework and therefore, we re-structured the whole manuscript. We added that this is a scoping review in the introduction. An introduction has been added and the table has been shortened and moved. To avoid repetitions and redundancy, many sections were shortened. We also added a section on conclusions and future directions and addressed the limitations.

Reviewer 3 Report

This article is on an important and as you point out an under-researched area.

Overall comment: This paper is far too long and appears repetitive. It is difficult to wade through it all and work out the main points being made. The conclusions are spot on but the arguments for the conclusion must be made more succinctly. Please carefully review each section and look to cut it by 50%.

Consideration of the welfare of assistance dogs (of any type) is incredibly important and this paper makes some very interesting points. But as I say above, they get lost in all the words.

The abstracts reads almost like an introduction but the paper does not really have an introduction. The abstract says the paper will consider three areas and then launches into the first one although it calls it the introduction. In the restructure please have an introduction which clearly states the plan for the article and what is going to be covered and what is not. Which countries will be included and why. And then have the sections. 

The current introduction starts with a section called the legal dimension but covers all sorts of other things and a little bit about the laws in the US. the second section in the introduction is called numbers and facts but covers more. Table 1 is too confusing and again tries to do too much. After 1.1 being called the legal dimension it has sections 1.3 and 1.4 which are the US and European legal dimensions respectively (seems like a double up).

Terminology is an important aspect and warrants coverage and this paper attempts to do it but it is confusing. A paper by Howell et al would be useful here (Animals 2022 https://doi.org/10.3390/ani12151975).

Please rewrite with a tighter focus.

Specific points: There are some incorrect word choices and some other English mistakes - not to many.

Please be consistent when using US. Either US or U.S. and define US the first time you use in the body of the paper. 

In section 3 perhaps start with info in 3.1.3 as many of the activities children undertake are mentioned several times. The section on children is good and appears novel. The discussion seems to roam over new ground so another area to think about where it should go and do I need so much.

I look forward to reading a shorted more organised paper.

Author Response

Reviewer 3

We appreciate the overall positive evaluation of the content of our manuscript and are pleased that the reviewer finds the topic interesting and timely. We are thankful for the careful evaluation of our manuscript. Your suggestions have been incorporated and helped us a lot to facilitate reading and to enhance the quality of our paper.

We agree that the structure of our initial submission has needed some rework and therefore, we re-structured the whole manuscript. An introduction has been added and the table has been shortened and moved. We added that this is a scoping review in the introduction. To avoid repetitions and redundancy, many sections were shortened, as you requested. We have included the suggested reference (Howell et al. 2022), which was really important to include. We also added a section on conclusions and future directions and addressed the limitations.

Round 2

Reviewer 1 Report

Thank you for addressing my comments made to your last submitted version. I am happy with the current version. Two minor issues: 1) Some readers may ask what keywords you used to find the relevant literature and information. A proper scoping review requires the researchers to report the search strategies. The authors may consider adding that as an Appendix to the manuscript. 2) Please read the manuscript carefully, there are still one or two spelling errors that I had identified. 

Author Response

Thank you again for reading the manuscript and providing suggestions for further improvement! The search and inclusion criteria for the literature have now been added to the text. Moreover, spelling and abbreviations have been checked (also in response to Reviewer #3).

Reviewer 2 Report

I appreciate the revisions made by the authors.

Author Response

Thank you again for reading the manuscript! 

Reviewer 3 Report

Thank you for the re-organisation of the manuscript which has substantially improved the paper. I only have a couple of minor changes to suggest.

You appear to use the US, America and North America interchangeably. Can you please be consistent. The US is obviously a country and North America a regions (including the US, Canada and Mexico) but America is unclear.

There are still a couple of places where an abbreviation is used but not given in full, e.g. ADI in line 107.

The section starting with line 428 about a study conducted in Australia - please introduce the study in the first sentence and include the reference there and then talk about the findings.

Author Response

Thank you again for reading the manuscript and providing suggestions for further improvement! Specifications on country or regions have been added and all abbreviations have been checked and corrected where needed. The study by Gibson & Oliva (2022) is now adequately cited in the paragraph.